# Viral Community and Novel Viral Genomes Associated with the Sugarcane Weevil, *Sphenophorus levis* (Coleoptera: Curculionidae) in Brazil

**DOI:** 10.3390/v17101312

**Published:** 2025-09-28

**Authors:** Amanda Haisi, Márcia Furlan Nogueira, Fábio Sossai Possebon, João Pessoa Araújo Junior, Jeanne Scardini Marinho-Prado

**Affiliations:** 1Biotechnology Institute, Sao Paulo State University (UNESP), Botucatu 18607-440, SP, Brazil; amanda.haisi@unesp.br (A.H.); fabio.possebon@unesp.br (F.S.P.); joao.pessoa@unesp.br (J.P.A.J.); 2Brazilian Agricultural Research Corporation-Embrapa Environment, Jaguariúna 13918-110, SP, Brazil; jeanne.marinho@embrapa.br; 3School of Veterinary Medicine and Animal Science, São Paulo State University (UNESP), Botucatu 18618-681, SP, Brazil

**Keywords:** virome, reo-like virus, tombus-like virus, picorna-like virus, biocontrol

## Abstract

*Sphenophorus levis*, commonly known as the sugarcane weevil, is one of the most important pests affecting Brazilian sugarcane crops. It has spread to all sugarcane-producing regions of Brazil, mainly through contaminated stalks. Effective control of this pest is difficult due to the protection conferred by the host plant during the larval stage. As a result, despite current control measures, *S. levis* populations continue to grow, and reports of new infestations remain frequent. Biotechnological control measures, such as the use of viruses, stands as a promising tool for pest control in agriculture. The aim of this study was to explore the RNA virome associated with *S. levis* using a viral metagenomic approach. Through the Read Annotation Tool (RAT) pipeline, we characterized, for the first time, the gut-associated viral community in adult weevils, identifying several novel viral genomes. *Sphenophorus levis*-associated virus (SLAV) had 12,414 nucleotides (nt); *Sphenophorus levis* tombus-like virus (SLTV) had 4085 nt; and the four genomic segments of *Sphenophorus levis* reo-like virus (SLRV) ranged from 2021 to 4386 nt. These genomes were assembled from 65,759 reads (SLAV), 114,441 reads (SLTV), and 270,384 reads (SLRV). Among the detected viral families, *Partitiviridae* was the most abundant. The identification of possible viral pathogens lays the foundation for future research into their potential use as biological control agents against *S. levis*.

## 1. Introduction

The genus *Sphenophorus* (Coleoptera: Curculionidae) is found in several countries across several continents, comprising a complex of species that are pests of essential crops within the grass group [1]. In North America, where the genus is believed to have originated, 75 species are recorded, 18 in South America, and six in Europe, North Africa, and Asia. Twenty-six species are recorded in the remaining areas of Africa and the Pacific region [2]. Fourteen species were described in Brazil, including *S. levis* Vaurie, 1978, with a range that also extends to Argentina and Uruguay [1,2].

*Sphenophorus levis* is known as the sugarcane weevil, one of the most important pests of Brazilian sugarcane crops [1,2], that has spread to all sugarcane-producing Brazilian regions via contaminated stalks [3]. The larvae of *S. levis* open galleries in the rhizomes, resulting in symptoms of leaf and tiller yellowing and drying. The damage is reflected in the number of final stalks for harvesting, and economic losses can be estimated based on the reduction in the expected number of tons of sugarcane [4]. Adults have a long lifespan and are primarily nocturnal insects, with most activities (walking, digging, and mating) occurring at night [5].

With the change in the Brazilian harvesting system from manual harvesting of burnt sugarcane to mechanical harvesting of green sugarcane, populations of *S. levis* rapidly increased. In some regions, the insects decimated the sugarcane fields within one or two harvests [3]. Controlling the sugarcane weevil is challenging because the insect is protected inside the plant during the larval period. Nowadays, it is recommended to mechanically destroy the stubble, establish a fallow period in the planting area, and use insecticides on the planting furrow and sugarcane stubble [1,6,7]. Despite all the control measures currently employed, the population of the weevil in sugarcane areas has still increased, with records of newly infested areas being frequent over the years [1,3].

In the USA and Japan, some important species of *Sphenophorus* are efficiently controlled using entomopathogenic nematodes [1]. In Brazil, studies indicate the use of the nematodes of the genera *Heterorhabditis* and *Steinernema* (Nematoda: Rhabditidae) as a viable alternative to control the larvae and pupae of *S. levis* inside rhizomes [1,8]. *Steinernema carpocapsae* was found to cause natural infection in *S. levis* inside the root, resulting in up to 60% pupal mortality. The application of *S. carpocapsae* just after cane harvesting is suggested to reduce the *S. levis* population in the next plant generation [8]. Entomopathogenic fungi can also be used to control *S. levis*. There are studies on *Metarhizium anisopliae*, *Beauveria bassiana*, and *B. caledonica*; however, their effectiveness in the field is not always consistent, and there is a lack of information on application methods, doses, and formulations [9,10,11].

In the reviewed literature, no studies were found on the RNA virome of *S. levis,* and thus potential viral biocontrol agents remain unidentified; however, some do describe viral families and species in other Curculionidae (Table 1, [12,13,14,15,16,17,18]). Experimental infections with Invertebrate iridescent virus 6 (IIV6) were conducted on *Anthonomus grandis*, the boll weevil [19] and *Diaprepes abbreviatus*, the citrus weevil [20]. Additionally, *Cypovirus* sp. was studied as a biocontrol agent for *Rhynchophorus ferrugineus*, the red palm weevil [21]. All studies reported productive infections.

The objective of this work was to explore the diversity of RNA viruses associated with the sugarcane weevil using a viral metagenomic approach. We characterized, for the first time, the intestinal viral community present in this pest, including novel putative viruses identified in adult weevils. The identification of possible viral pathogens is the first step in investigating their potential as biological control agents.

## 2. Materials and Methods

### 2.1. Insect Collection

Adult specimens of *Sphenophorus levis* were collected in March 2023 from sugarcane plantations in the city of São Joaquim da Barra, SP (−20.48266667 S, −47.87330556 W) (Figure 1), using baits made from sugarcane billets. Groups of live insects collected were placed in plastic containers (14.5 × 10.0 × 5.0 cm) with perforated lids and transported by motor vehicle under controlled conditions (25 °C, complete darkness) for 3 h 30 min to the Laboratory of Entomology and Phytopathology (LEF) of Embrapa Environment (CNPMA) in Jaguariúna, SP. Upon arrival, 30 individuals were randomly selected and fasted for 72 h in a BOD incubator maintained at 25 °C with a 24 h scotophase. The intestines of these insects were then removed and immediately stored in RNAlater (Sigma-Aldrich, St. Louis, MO, USA), resulting in 3 pools of 10 intestines each [22]. Samples were frozen at −20 °C, following the manufacturer’s instructions, until RNA extraction.

### 2.2. Total RNA Extraction, Library Preparation, and Next-Generation Sequencing (NGS)

At the Institute of Biotechnology (IBTEC) of UNESP in Botucatu, SP, total RNA was extracted from each pool of intestines using a Cell Disruptor (Loccus, Cotia, Sao Paulo, Brazil) and the Total RNA Purification Kit (Norgen Biotek Corp., Thorold, ON, Canada). Total RNA was quantified with the Qubit RNA High Sensitivity (HS) Assay Kit (Thermo Fisher Scientific, Waltham, MA, USA) and assessed for integrity using the High Sensitivity RNA ScreenTape System (Agilent Technologies, Santa Clara, CA, USA). One aliquot of total RNA from each sample was processed with the NEBNext Poly(A) mRNA Magnetic Isolation Module (New England Biolabs Inc., Ipswich, MA, USA) for mRNA isolation. Six RNA-seq libraries were constructed from total RNA (3) and mRNA (3) using the Zymo-Seq RiboFree Total RNA Library Kit (Zymo Research, Irvine, CA, USA) and sequenced paired-end using the NextSeq 500/550 High Output Kit v2.5 (150 Cycles) (Illumina) on the NextSeq 500 system (Illumina, San Diego, CA, USA). All reagents were used according to the manufacturers’ instructions. To monitor the potential contamination during the workflow, a negative control consisting of a pool of *Cosmopolites sordidus* (banana weevil) specimens was processed in parallel through all steps of nucleic acid extraction, library preparation, and sequencing.

### 2.3. RNA-Seq Data Analysis

The obtained sequences were subjected to quality control using the programs FastQC and MultiQC (https://github.com/s-andrews/FastQC. Accessed on 13 July 2023). Trimmomatic v0.39 [23] was used to remove adapter sequences and low-quality reads. High-quality reads were used for de novo assembly using SPAdes 3.15.2 [24] for RNA sequences [25]. The resulting contigs were subjected to taxonomic classification and relative abundance calculation using the Read Annotation Tool (RAT) pipeline [26], utilizing the NCBI non-redundant protein database (nr) (version 2023-11-20). The data obtained were tabulated in Excel for the preparation of tables and figures. Rarefaction curve analyses were performed using the vegan R package v.2.7.1 in R [27]. Curves were computed for each library with 100 resampling steps, and richness values (viral families) were plotted as mean ± standard error (SE).

### 2.4. Viral Genome Analysis

Based on the RAT results, contigs with the highest relative abundance—each supported by more than 200,000 mapped reads and classified as “not assigned (NA)”—were selected for viral genome assembly. Contigs with minimum threshold of ≥500 nucleotides and an average coverage of ≥20× were aligned using Geneious Prime v. 2025.1.3 (Dotmatics, Boston, MA, USA), and the longest contigs were subsequently used for molecular characterization of putative novel viral genomes. Subsequently, putative Open Reading Frames (ORFs) were identified using ORF finder (https://www.ncbi.nlm.nih.gov/orffinder/. Accessed on 11 November 2024) with a length cutoff of >300 nt. The conserved domains were predicted using InterProScan (http://www.ebi.ac.uk/interpro. Accessed on 11 November 2024) [28] on amino acid sequence, as well as Conserved Domain Database (CDD) v. 3.21 and Pfam v.35 on nucleotide sequences, both with a significance threshold of E-value ≤ 10^−3^. The classified contigs were subjected to BLASTx analysis against the non-redundant protein database (nr) using an e-value cutoff of <10^−5^ to eliminate likely false positives. Potential novel genomes were further assessed based on their genomic organization and phylogenetic topology.

### 2.5. Phylogenetic Analysis

In order to obtain more insightful information about the classification of the putative novel viruses, the amino acid sequences of the complete polyprotein or sequences that exhibiting similarity to RNA polymerase were used to construct phylogenetic trees. For each set of sequences, a global alignment was performed using MAFFT v7.520 [29], while phylogenetic tree construction was conducted with IQ-TREE 2 v2.2.0.3 [30] using ModelFinder parameter for selecting the best statistical model. Bootstrap analysis was performed with 1000 replications to assess the robustness of the tree. To visualize and edit the resulting trees, the software FigTree v1.4.4 (http://tree.bio.ed.ac.uk/software/figtree. Accessed on 15 November 2024) was used.

## 3. Results and Discussion

The number of clean reads obtained from the 3 samples analyzed after the removal of low-quality reads and adapters ranged from 64,836,018 to 73,325,198 (Table 2). To improve the chance of viral detection, two RNA sequencing methods were used, mRNA and Total RNA. Of the total number of reads, 205,722,814 were from mRNA sequencing and 201,941,280 from total RNA. The total number of assembled contigs and scaffolds ranged from 49,708 to 89,845, with 151,588 (36.30%) originating from mRNA and 265,960 (63.70%) from total RNA. The number of contigs classified as viruses, 239, was also lower in the mRNA samples than in the total RNA samples, totaling 39 (16.32%) and 200 (83.68%) contigs, respectively. The number of reads from the contigs classified as viral, 1,907,762, corresponds to 0.47% of the total clean reads, and of these, approximately half were assigned a taxonomic category.

### 3.1. Virome Composition

Among those classified as viral, 26 contigs, with sizes ranging from 232 to 5256 bp, corresponding to 930,951 (48.80%) of the “viral” reads, could not be identified by RAT at any taxonomic level, which reaffirms the fact that very little is known about insect viruses. Of the remaining 213 contigs, those resulting from 963,752 reads were assigned to 10 families and six putative species of RNA viruses (98.66%), and from 13,059 reads to five families and three putative species of DNA viruses (1.34%) (Figure 2 and Figure 3, and Table 3 and Appendix A). As shown by the rarefaction curves of the three samples reaching the asymptote (Appendix A), the sequencing depth was sufficient to capture the viral community at family level. The family *Partitiviridae* was observed in samples 5 and 6, *Tombusviridae* in sample 7, *Virgaviridae* and *Benyviridae* in sample 6, and *Nudiviridae* in sample 5 only. The remaining 10 families occurred in all samples. The presence of reads attributed to DNA viruses may be due to the sequencing of both residual DNA and mRNA in the total RNA extracted from the samples.

Despite occurring in two of the three samples analyzed, the highest abundance of reads among those categorized into families was classified as *Partitiviridae*, which corroborates the findings for the alfalfa weevil *Hypera postica* (Coleoptera: Curculionidae), where only *Iflaviridae* was more abundant [17]. *Partitiviridae* are associated with latent infections of their fungal, protozoan, and plant hosts, and there are no known natural vectors [31]. However, among the unclassified *Partitiviridae*, there are those observed in insects. A high viral abundance, represented by 11,419 reads, was detected in the total RNA library from sample 6 and was classified by RAT as homologous to *Drosophila biauraria* male-killing partitivirus 1 (Appendix A), a maternally inherited virus belonging to the family *Partitiviridae* (designated DbMKPV1) that was reported to induce male-killing in *Drosophila* (Diptera: Drosophilidae) [42]. *Totiviridae* are also dsRNA viruses and similarly associated with latent infections of their fungal or protozoan hosts.

Among the ssRNA(−) viruses, *Chuviridae* and *Aliusviridae* belong to the order *Jingchuvirales* [33,43] and, together with *Phasmaviridae*, are families of insect viruses observed in the total RNA of the three studied samples. *Chuviridae* was discovered using viral metagenomics, and the virion is unknown, presumably non-enveloped. Viral sequences homologous to the unclassified Coleopteran chu-related virus OKIAV151 were obtained from total RNA libraries, with a total of 4608 reads mapped (Appendix A). This virus was also observed in *Larinus minutus* (Coleoptera: Curculionidae) [13], a weevil that has been released in the USA as part of a biological control program to manage spotted and diffuse knapweed (*Centaurea maculosa* Lam. and *C. diffusa* Lam.) [44]. *Phasmaviridae* are viruses maintained in and/or transmitted by blattodean, coleopteran, dipteran, hemipteran, hymenopteran, neuropteran, and odonatan insects [34]. Viral abundances were assigned to the family *Phasmaviridae,* with 2106 reads mapped from total RNA libraries. Among these, 1339 reads, detected exclusively in samples 5 and 6, exhibit homology to the genus *Orthophasmavirus* (Appendix A).

*Orthomyxoviridae* and *Rhabdoviridae* are also ssRNA(−). The former have a wide range of hosts, including invertebrates, and are associated with acute febrile respiratory tract infections and zoonosis. Viral abundances were assigned to the family *Orthomyxoviridae* with 1279 reads mapped, including 1137 reads showing similarity to Coleopteran orthomyxo-related virus OKIAV200 (Appendix A). This virus was also found in the transcriptome of the weevil *Ips typographus* (Coleoptera: Curculionidae) [13], which is one of the most economically important species associated with spruce in Central Europe [12]. The family *Rhabdoviridae* is ecologically diverse, with members infecting plants or animals, including mammals, birds, reptiles, amphibians, or fish. Rhabdoviruses are also detected in invertebrates, including arthropods, some of which may serve as unique hosts or may act as biological vectors for transmission to other animals or plants [35].

*Tombusviridae*, *Virgaviridae*, and *Benyviridae* are ssRNA(+) families of plant viruses. Numerous tombus-related viruses have recently been described in non-plant hosts, such as marine invertebrates and terrestrial arthropods [45]. These viruses exhibit a high degree of genetic diversity and branch basal to the plant-associated *Tombusviridae* [46] CPMoV (cowpea mottle virus), bean mild mosaic virus, and TCV (turnip crinkle virus) are transmitted by beetles (Coleoptera) [32]. Three distinct tombus-like viruses—*Ips* tombus-like viruses 1, 2, and 3,—two different virga-like viruses—*Ips* virga-like viruses 1 and 2—and one partial sequence affiliated with beny-like viruses, named *Ips* beny-like virus 1, were identified in the RNA-Seq library of the European spruce bark beetle, *Ips typographus* (Coleoptera: Curculionidae) [14].

Endogenous viral elements (EVEs) are viral genomes or fragments of viral genomes of non-retroviral origin that are integrated into the genomes of their hosts. The currently known insect EVEs belong to at least 28 viral families [47]. *Polydnaviriformidae* are viruses of parasitic Lepidoptera wasps (Ichneumonidae and Braconidae), which are vertically transmitted to their offspring. The wasp injects one or more eggs into its host along with a quantity of virus which does not replicate (non-replicative host) but the expression of viral genes prevents its immune system from killing the wasp’s egg and causes other physiological alterations that ultimately cause the parasitized host to die [48]. From the total RNA and mRNA enrichment of the three samples analyzed, 5368 reads were attributed to the *Bracoviriform facetosae.*

*Eupolintoviridae* is a new family name for Polintons, also known as Mavericks, connoting similarities to adenovirus virion proteins and the presence of a retrovirus-like integrase gene. Many polintons encode possible capsid proteins and viral genome-packaging ATPases, supporting the inference that at least some polintons are viruses capable of cell-to-cell spread [49]. Of the 2363 reads identified as belonging to the family *Eupolintoviridae*, 2073 were attributed to Drosophila-associated adintovirus 2 and were detected exclusively in total RNA libraries. Extended bracovirus sequences, including *Bracoviriform facetosae*, were also found in the reference genome of the Colorado potato beetle, *Leptinotarsa decemlineata* (Coleoptera: Chrysomelidae), and in the genomes of several other representatives of Coleoptera. The presence of bracovirus sequences in the genetic material isolated from both imago and larval insect tissues, as well as from sterile eggs of *L. decemlineata*, was demonstrated, suggesting the integration of bracovirus genomic sequences within the Colorado potato beetle genome. Additionally, in this Coleoptera, more than 1000 contigs, primarily obtained from genomic data, were annotated as belonging to different members of the family *Eupolintoviridae*. The alignments indicate homology not only in terms of retrovirus-like integrase and polymerase B but also in terms of structural proteins of the family *Eupolintoviridae* [38,50]. Drosophila-associated adintovirus 2 was also described in *Drosophila* collected in Austria [51] and observed in sequences derived from bat feces [52].

Although reads were also mapped to other double-stranded DNA virus families, their abundances were considerably lower than *Polydnaviriformidae* and *Eupolintoviridae*. Among these were *Iridoviridae* and *Nudiviridae*, which infect invertebrates, represented by 348 and 121 mapped reads, respectively. Members of both families have been studied for biological control purposes. Invertebrate Iridescent Virus 6 (IIV6), also referred to as Chilo Iridescent Virus (CIV), has been studied as a potential biological control agent, including for the boll weevil, *Anthonomus grandis* (Coleoptera: Curculionidae), and for *Phyllophaga vandinei* (Coleoptera: Scarabaeidae), a pest of tropical fruit trees [19,53,54]. The Oryctes Rhinoceros Nudivirus (OrNV), in turn, has been successfully used in the control of the Coconut rhinoceros beetle *Oryctes rhinoceros* L. (Coleoptera: Dynastidae), one of the major pests of coconut and oil palms in the Asia-Pacific region [55].

Of the 976,811 reads assigned by RAT to RNA and DNA virus families based on highest similarity, approximately 0.1% (981) were classified as belonging to ssDNA virus, including members of the family *Parvoviridae*. The two subfamilies, *Parvovirinae* and *Densovirinae*, are distinguished primarily by their respective ability to infect vertebrates (including humans) versus invertebrates. An outbreak of densovirus with high mortality was recently described in a commercial farm of *Tenebrio molitor* (Coleoptera: Tenebrionidae) [56].

### 3.2. Characterization of Novel Viral Genomes

#### 3.2.1. Sphenophorus Levis Associated Virus (SLAV)

The deduced viral genome of a picorna-like virus, provisionally named *Sphenophorus levis* associated virus (SLAV), is 12,414 nt long and was assembled from 65,754 reads from sample 5 (total RNA library), showing a mean coverage of 381× and a maximum coverage of 1853×. The BLASTx analysis showed 32.62% of identity (Query Cover: 68%; *E. value*: 0.0) with *Hypera postica* associated virus 1 (Genbank accession number: MW676138), which was described in *Hypera postica* (Coleoptera: Curculionidae), a weevil that primarily feeds on alfalfa (*Medicago sativa*) [17]. In total, 17 additional contigs ranging from 170 to 11,578 bp were recovered from total RNA and mRNA enriched libraries from all samples, showing 98.8% and 100% of identity with SLAV.

The SLAV genome included a single open reading frame (ORF) encoding a 4071 amino acid product. Conserved structural domains typically found in members of the *Picornavirales* order were identified, including the RNA helicase, RNA-dependent RNA polymerase (RdRp), protease and picornavirus capsid protein domain-like (RHV) (Figure 4A).

To investigate the genetic relationships between SLAV and other arthropods and plant viruses belonging to the *Picornavirales* order, a phylogenetic tree was constructed based on the complete coding region of SLAV and 44 sequences from viruses belonging to the families *Solinviviridae*, *Secoviridae*, *Iflaviridae,* and unclassified picorna-like viruses. Phylogenetic analyses based on amino acid sequences indicated that SLAV has a close relationship with unclassified ssRNA viruses, such as *Hypera postica* associated virus 1 (Genbank accession number: MW676138) [17] and *Diabrotica undecimpunctata* virus 1 (Genbank accession number: MN646770), found in that spotted cucumber beetle [57] forming an unclassified group closely related to members of the family *Solinviviridae* (Figure 4B). It is also included in this last family the *Diabrotica virgivera virgivera* virus 2, identified in the Western corn rootworm *Diabrotica virgifera virgifera* LeConte (Coleoptera: Chrysomelidae), known to cause severe economic losses in maize.

Viruses belonging to the order *Picornavirales* are frequently described in arthropods, with picorna/calici-like viruses representing a significant proportion of the viral reads detected in viromes of the class Insecta [45], as observed for SLAV, which total RNA libraries showed a relative abundance of reads mapped ranging from 28,654 to 66,987.

#### 3.2.2. *Sphenophorus levis* Reo-like Virus (SLRV)

Two monocistronic genome segments with similarity to VP1 and VP2 proteins of a member of the order *Reovirales* were found by RAT in the total RNA libraries of all three samples. Through manual curation of unclassified contigs at the “viruses” superkingdom level, as well as from total RNA libraries, we identified two additional putative reovirus genome segments that likely encode VP6 and VP7 proteins. Pairwise nucleotide identity analysis showed that the segments found in different pools are highly similar—99.4% (4354 bp and 4377 bp), 99.3% (3279 and 3885 bp), 98.7% (2162 bp and 2147 bp) and 99.04% (1984 bp and 2017 bp). These correspond to the largest contigs encoding VP1 (4361 bp), VP2 (3907 bp), VP6 (2171 bp), and VP7 (2021 bp), respectively; therefore, the latter contigs (Figure 5A) were used to characterize the novel viral genome detected in the RNA library.

The contig homologous to VP1 was assembled from 216,239 reads mapped from sample 6 (total RNA library), showing a mean coverage of 4386× and a maximum coverage of 14,946×. In contrast, the longest contigs with homology to VP2, VP6 and VP7 were assembled from sample 7 (total RNA library). The putative VP2 segment was assembled from 99,596 mapped reads, with a mean coverage of 1846× and a maximum coverage of 5746. The putative VP6 segment was assembled from 49,029 reads mapped with a mean coverage of 1648× and a maximum coverage of 6172×. Finally, the putative VP7 segment was assembled from 56,005 reads with a mean coverage of 1648× and a maximum coverage of 9393× (Figure 5A).

The BLASTx analysis of all viral segments revealed similarity with the Berke-Baary Melophagus reo-like virus, previously described in the sheep ectoparasite *Melophagus ovinus* (Diptera: Hippoboscidae) from Russia [58]. The percentage of amino acid identity was 34.06% (Genbank accession number: UJG27935.1, Query Cover: 82%, E-value: 9 × 10^−27^) and 30.97% (Genbank accession number: UJG27936.1 Query Cover: 86%, E-value: 7 × 10^−163^) for VP1 and VP2, respectively. VP6 and VP7 showed 23.74% (Genbank accession number: UJG27940.1 Query Cover: 85%, E-value: 00) and 30.7% (Genbank accession number: UJG27941.1, Query Cover: 82%, E-value: 2 × 10^−39^) of amino acid identity, respectively.

The partial genome of the provisionally named *Sphenophorus levis* reo-like virus (SLRV), comprising four genomic segments, belongs to the order *Reovirales*. To investigate the genetic relationship among SLRV and other reoviruses, a phylogenetic tree was constructed based on the amino acid sequence of the putative RdRp segment, using 54 prototype sequences (Figure 5B). The phylogenetic analysis demonstrated that SLRV forms a divergent branch in *Spinareoviridae* family, closely related to the clade composed of unclassified reoviruses identified in Diptera, such as Hubei diptera virus 20 (Genbank accession number: KX884693) [45] and *Ceratitis capitata* reo-like virus 1 (Genbank accession number: OL957310) [59], as well as reoviruses identified in animal samples, including Bloomfield virus, isolated from fecal samples of wild mice (Genbank accession number: MF416371) [60], and the Burke-Barry Melophagus reo-like virus (Genbank accession number: OL420682 and OL420692) [58] already mentioned. Additionally, the unclassified reoviruses group appears to form a sister clade to the genus *Fijivirus*.

Currently, the family *Spinareoviridae* comprises nine genera and 58 species recognized by the ICTV [61]; however, novel viral genomes have been frequently described in metagenomic studies [62], and many of them remain unclassified, highlighting an as-yet unknown diversity within the order *Reovirales*. Regarding members of the genus *Fijivirus*, only the species *Nilaparvata lugens reovirus*—NLRV (*Fijivirus nilaparvatae*) is recognized as replicating exclusively in insects [63]. Other members of the genus can infect plants, with insects serving as the main vectors of infection [64]. Although we have not determined all the segments of SLRV, its average GC content was 41.8%, which differs from that observed in members of the *Fijivirus* genus, whose GC content ranges from 34% to 36% [64].

#### 3.2.3. *Sphenophorus levis* Tombus-like Virus (SLTV)

The deduced viral genome provisionally named *Sphenophorus levis* tombus-like virus (SLTV) belongs to the order *Tolivirales*. It is 4065 nt in length and was assembled from 114,441 reads obtained from sample 6 (Total RNA seq 6), with an average coverage of 1955× and a maximum coverage of 9123×. Additional contigs from samples Total RNA seq 5 and Total RNA seq 7 (ranging from 1657 to 4085 bp) were also classified as *Riboviria* sp. and, due to a pairwise identity of 99.1% when compared to SLTV, were considered to represent the same viral genome.

The SLTV genome contains four open reading frames (ORFs): ORF1 (954 bp; 317 amino acids), ORF2 (1548 bp; 515 aa), ORF3 (771 bp; 256 aa), and ORF4 (672 bp; 223 aa) (Figure 6A). ORFs 1 and 4 encode hypothetical proteins with no detectable similarity to known proteins in BLASTx. The BLASTx analysis of ORF2 revealed 64.67% identity to *Riboviria* sp. (Genbank accession number: WKV34284.1, Query Cover: 97%, E-value: 0.0) identified in a bird metagenome [65], and 52.33% identity to Soybean thrips tombus-like virus 3 (Genbank accession number: QP18784.2, Query Cover: 90%, E-value: 1 × 10^−161^). The ORF3 also showed 39.33% identity to Soybean thrips tombus-like virus 5 (Genbank accession number: QQP18796.1, Query Cover: 58%, E-value: 3 × 10^−34^), both detected in *Neohydatothrips variabilis* collected in the USA in 2018 [66]. It also showed 42% identity to Coleopteran tombus-related virus (Genbank accession number: QTJ63624.1 Query Cover: 58%, E-value: 1 × 10^33^) described in *Ips typographus* collected in Germany in 2012 [46].

Conserved domains were found, the ORF2 encodes the catalytic core domain of the RNA-dependent RNA polymerase (RdRp) from the *Calvusvirinae* subfamily (cd23234; E-value: 1.25 × 10^−42^). However, viruses belonging to the genus *Umbravirus*—the only genus described so far within this subfamily—do not encode a coat protein (CP) [67], in contrast to what is observed in SLTV, whose ORF3 encodes a capsid protein containing the S domain (PF00729; E. value: 3.12 × 10^−07^), which is shared by a wide range of viral capsid proteins, including those of the genus *Tombusvirus*. Although the overall genome organization of SLTV resembles that of members of the *Tombusviridae* family, no evidence was found for a translational readthrough mechanism or ribosomal frameshifting that would result in a −1 reading frame shift between ORF1 and ORF2, or for the presence of amber stop codons in ORF1 that would lead to the expression of a larger protein encoding the RdRp [32]

The genome of SLTV contains regions with overlapping ORFs, similar to tombus-like viruses described in insect transcriptomes [46]. The phylogenetic analysis based on amino acid sequences of RdRp domain (Figure 6B) demonstrated the close relationship between *Sphenophorus levis* tombus-like virus and unclassified members of *Tombusviridae*, such as Soybean thrips tombus virus 3 (Genbank acession number: MT240790) detected in *Neohydatothrips variabilis* (Thysanoptera: Thripidae), as well as Hymenopteran tombus-related virus (Genbank accession number: MW208794) described in the wasp *Alastor atropos* (Hymenoptera: Eumenidae) [46]. The clade also includes *Sclerotinia sclerotiorum* umbra-like virus 1 (Genbank accession number: NC_030203), an unclassified umbravirus described in a fungal plant, the white mold *Sclerotinia sclerotiorum* [68] (Figure 6B).

## 4. Conclusions

This study revealed, for the first time, the viral community associated with the sugarcane weevil, a pest of great economic importance to Brazilian sugarcane crops. The discovery of novel viral genomes, along with the high viral abundance, suggests the possibility of viral replication in these hosts and highlights an as-yet-undescribed viral diversity associated with these curculionids. These studies reveal promising avenues for identifying potential biological control agents targeting *Sphenophorus levis*. However, further studies are needed to assess the diversity of the viral community in different populations of *S. levis* and throughout its developmental stages. Still, continued research is essential to investigate the potential and applicability of these viruses in integrated pest management strategies.

## Figures and Tables

**Figure 1 viruses-17-01312-f001:**
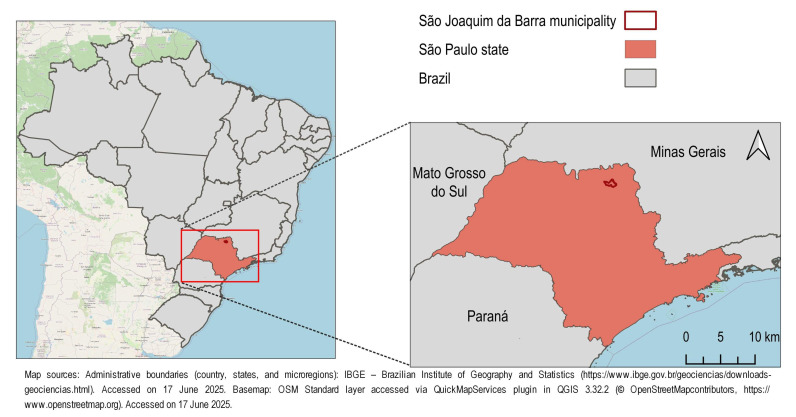
Sampling location of *Sphenophorus levis* adults. Specimens were collected in March 2023 from sugarcane plantations in the municipality of São Joaquim da Barra, São Paulo state, Brazil.

**Figure 2 viruses-17-01312-f002:**
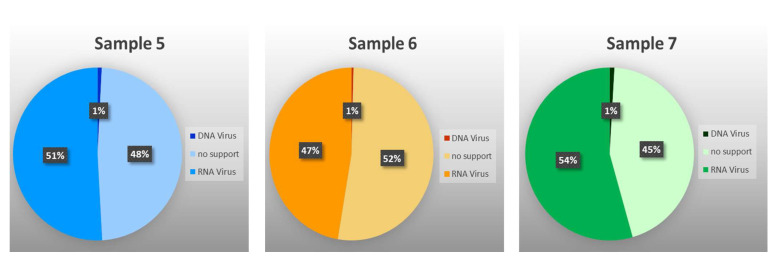
Percentage of reads either unclassified or assigned to RNA or DNA virus families or species.

**Figure 3 viruses-17-01312-f003:**
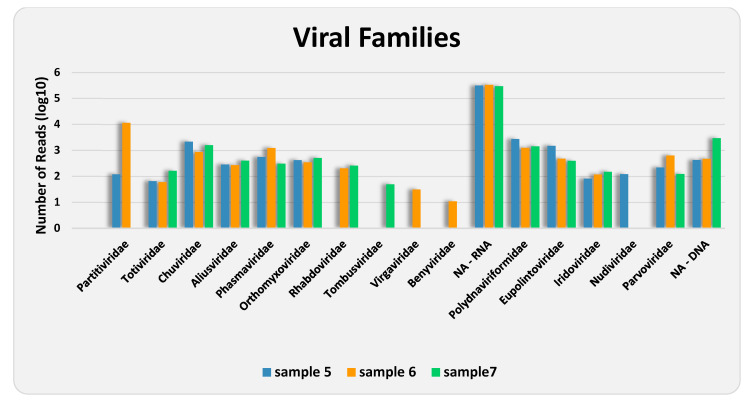
Abundance of reads either assigned or not assigned (NA) to various viral families, in both treatments of each analyzed sample.

**Figure 4 viruses-17-01312-f004:**
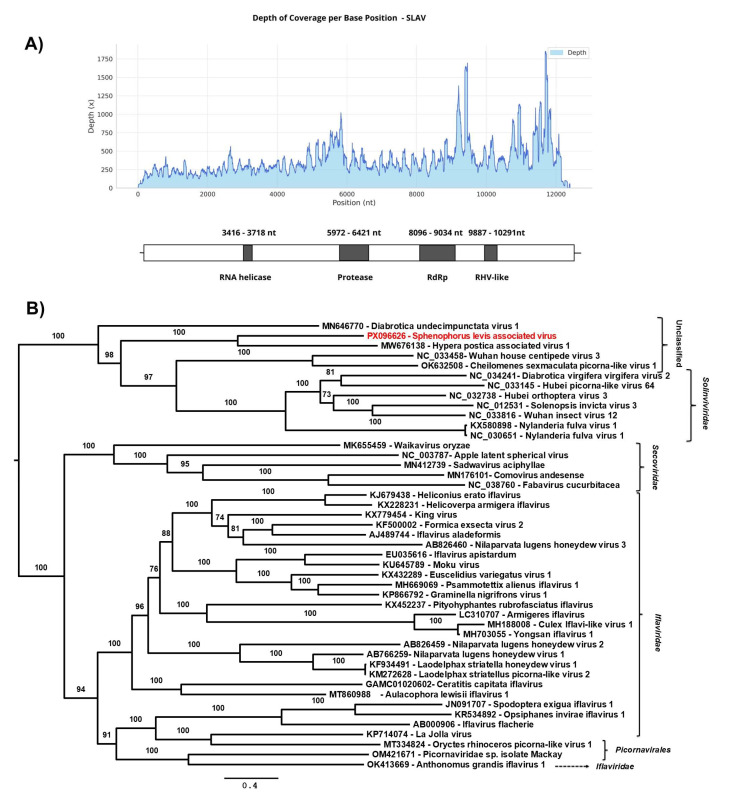
(**A**) Graph of depth coverage per base position and genetic structure of *Sphenophorus levis* associated virus. RNA helicase, RdRp; RNA-dependent RNA polymerase, RHV-like; (**B**) The maximum likelihood phylogenetic tree of *Sphenophorus levis*-associated virus and previously reported members of the order *Picornavirales* based on amino acid sequences of polyprotein. The phylogenetic tree was reconstructed using the best-fit model chosen according to BIC (Q.fam + F + I + I + R5) determined by ModelFinder. The bootstrap analysis consisted of 1000 replicates, and the virus reported in this study is marked in red. Scale bar represents the number of amino acid substitutions per site.

**Figure 5 viruses-17-01312-f005:**
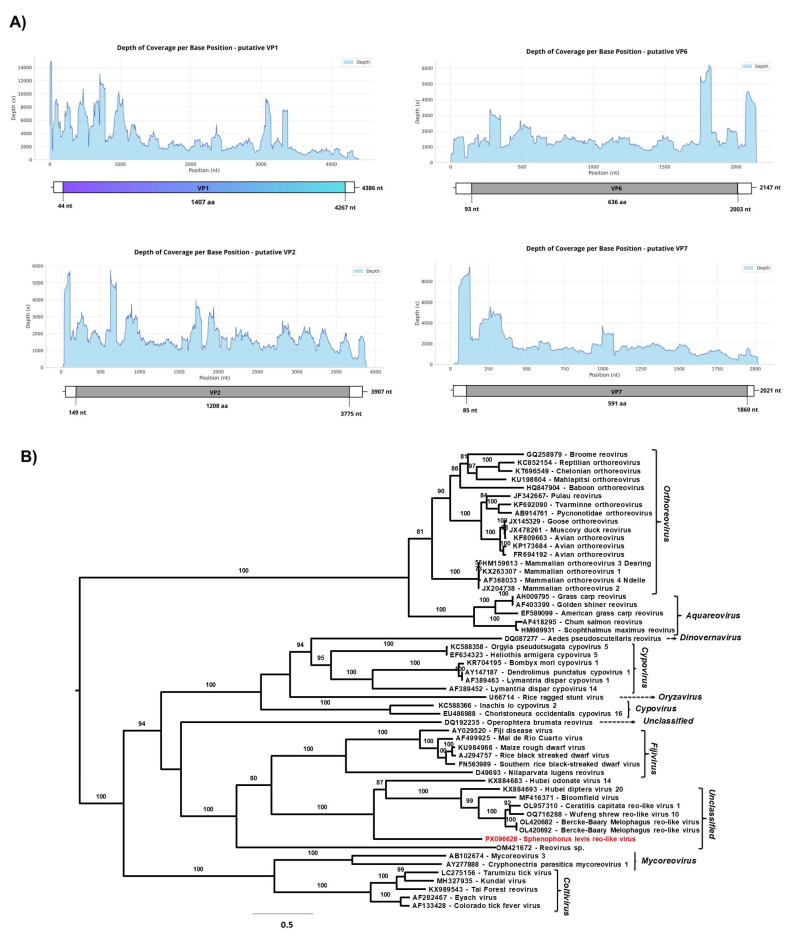
(**A**) Graph of depth coverage per base position and genomic organization of putative VP1, VP2, VP6, and VP7 genomic segments from *Sphenophorus levis* reo-like virus (**B**) The maximum likelihood phylogenetic tree of *Sphenophorus levis* reo-like virus and previously reported members of the *Spinareoviridae* family based on amino acid sequences of RdRp. The phylogenetic tree was reconstructed using the best-fit model chosen according to BIC (Q.fam + F + I + I + R5) determined by ModelFinder. The bootstrap analysis consisted of 1000 replicates, and the virus reported in this study is marked in red. The scale bar represents the number of amino acid substitutions per site.

**Figure 6 viruses-17-01312-f006:**
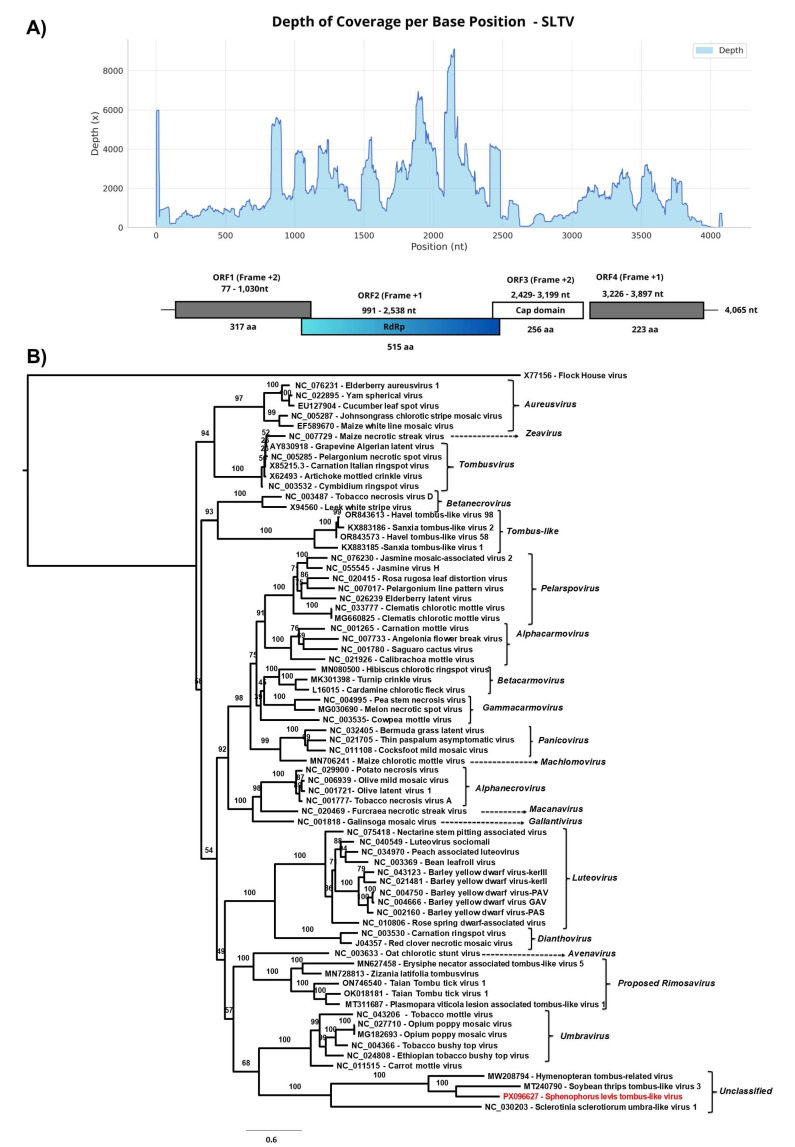
(**A**) Graph of depth coverage per base position and genetic structure of *Sphenophorus levis* tombus-like virus. (**B**) The maximum likelihood phylogenetic tree of *Sphenophorus levis* tombus-like virus and previously reported members of the *Tombusviridae* family based on amino acid sequences of RdRp. The Flock House virus was used as an outgroup. The phylogenetic tree was reconstructed using the best-fit model chosen according to BIC (LG + I + G) determined by ModelFinder. The bootstrap analysis consisted of 1000 replicates and the virus reported in this study is marked in red. Scale bar represents the number of amino acid substitutions per site.

**Table 1 viruses-17-01312-t001:** Viral families and species in other Curculionidae.

Host Common Name	Scientific Name	Location	Virus Name	Taxonomy	Accession	Ref.
European spruce bark beetle	*Ips typographus*	Czech Republic	Ips typographus entomopoxvirus (ItEPV)	*Poxviridae*	NA	[12]
		Germany	Coleopteran orthomyxo-related virus OKIAV196	*Orthomyxoviridae*	PRJNA183205	[13]
			Coleopteran orthomyxo-related virus OKIAV200			
		Finland	Ips virga-like virus 1 and 2	*Virgaviridae*	OR537183, OR537184	[14]
			Ips tombus-like virus 1, 2 and 3	*Tombusviridae*	OR537185, OR537186, OR537211	
			Ips spici-like virus 1	*Spiciviridae*	OR537187	
			Ips narna-like virus 1 and 2	*Narnaviridae*	OR537188, OR537189	
			Ips partiti-like virus 1	*Partitiviridae*	OR537190 to OR537193	
			Ips sobemo-like virus 1	*Solemoviridae*	OR537194, OR537195	
			Ips phenui-like virus 1 and 2	*Phenuiviridae*	OR537196, OR537198	
			Ips phenuiviral-like M segment 1 and 2		OR537197, OR537199	
			Ips phenuiviral-like M segment 2		OR537200	
			Ips erranti-like virus 1 to 6	*Metaviridae*	OR537201 to OR537206	
			Ips quenya-like virus 1	NA	OR537207 to OR537209	
			Ips beny-like virus 1	*Benyviridae*	OR537210	
Lesser knapweed flower weevil	*Larinus minutus*	USA	Coleopteran chu-related virus OKIAV151	*Chuviridae*	PRJNA183205	[13]
			Coleopteran phenui-related virus OKIAV293	*Phenuiviridae*		
Maize weevil	*Sitophilus zeamais*		Coleopteran phenui-related virus OKIAV287			
			Coleopteran orthomyxo-related virus OKIAV158	*Orthomyxoviridae*		
Citrus root weevil	*Diaprepes abbreviatus*		Coleopteran hanta-related virus OKIAV221	*Hantaviridae*		
Eucalyptus snout beetle	*Gonipterus* spp.	Brazil	Gonipterus platensis bunya-Like virus (GPV)	NA	MT435497, MT435498	[15]
			Gonipterus platensis macula-like virus (GPMV)	*Tymoviridae*	MT435496	
Rice weevil	*Sitophilus oryzae*	China	Weevil wasp positive-strand RNA virus 2 (WWPSRV-2)	*Iflaviridae*	MW864601	[16]
Alfalfa weevil	*Hypera postica*	France	Hypera postica associated alphaflexivirus (HpaAV)	*Alphaflexiviridae*	MW676130	[17]
			Hypera postica associated iflavirus 1 (HpaIV1)	*Iflaviridae*	MW676131	
			Hypera postica associated iflavirus 2 (HpaIV2)		MW676132	
			Hypera postica associated permutotetravirus	*Permutotetraviridae*	MW676133	
			Hypera postica associated sinaivirus	*Sinhaliviridae*	MW676134	
			Hypera postica associated sobemovirus 1	*Solemoviridae*	MW676135	
			Hypera postica associated sobemovirus 2		MW676136	
			Hypera postica associated sobemovirus 3		MW676137	
			Hypera postica associated virus 1 (HpaV1)	NA	MW676138	
Cotton boll weevil	*Anthonomus grandis*	Brazil	Anthonomus grandis iflavirus 1 (AgIV-1)	*Iflaviridae*	OK413669	[18]

NA = not assigned.

**Table 2 viruses-17-01312-t002:** Summary statistics of RNA-Seq libraries, de novo assembly and taxonomic classification by the RAT.

Sample_ Method	Clean Reads	Assembly Overview					Classified as Virus			
		n° Total Contigs	n° Contigs ≥ 100	n° Contigs ≥ 500	N50	L50	n° of Contigs	Contig Length Range (bp)	Reads	Family/Species Level (%)
Slevis5_ polyA	66,576,852	49,708	45,113	16,515	1711	4237	12	223–4944	31,708	31,708 (100)
Slevis5_ RNAtotal	64,836,018	88,189	78,445	20,910	1647	5040	69	98–12,414	593,718	291,497 (49.1)
Slevis6_ polyA	65,820,764	52,098	46,845	16,879	1754	4271	17	170–5513	20,403	20,136 (98.7)
Slevis6_ RNAtotal	66,187,054	87,926	77,964	20,527	1655	4920	60	207–11,578	712,046	329,353 (46.3)
Slevis7_ polyA	73,325,198	49,782	45,249	16,530	1707	4245	10	257–2552	29,589	29,589 (100)
Slevis7_ RNAtotal	70,918,208	89,845	79,265	20,892	1651	5008	71	73–6746	520,298	274,528 (52.8)
Total	407,664,094	417,548	372,881	112,253			239		1,907,762	976,811 (51.2)

**Table 3 viruses-17-01312-t003:** Characteristics of the genome and hosts of the viral families observed in *S. levis*.

Family	Genome	Genome Size	Host Range
*Partitiviridae* [31]	linear bipartite dsRNA	3–4.8 kbp	Plants, fungi and protozoa
*Totiviridae* [32]	linear monopartite dsRNA	4.6–6.7 kbp	Fungi and protozoa
*Chuviridae* [33]	linear or circular, monopartite or bipartite ssRNA(−)	9.1–12.2 kb	Arachnids, barnacles, crustaceans, insects, fish and reptiles
*Aliusviridae* [33]	linear monopartite ssRNA(−)	9.9–15.3 kb	Insects
*Phasmaviridae* [34]	linear tripartite ssRNA(−)	9.7–15.8 kb	Insects
*Orthomyxoviridae* [32]	linear multipartite ssRNA(−)	~13.5 kb	Aquatic birds, human, pig, horse and seals
*Rhabdoviridae* [35]	linear monopartite or bipartite ssRNA(−)	10–16 kb	Vertebrates, invertebrates and plants
*Tombusviridae* [32]	linear monopartite or bipartite ssRNA(+)	3.7–4.8 kb	Plants
*Virgaviridae* [36]	linear monopartite or multipartite ssRNA(+)	6.3–13 kb	Plants
*Benyviridae* [37]	linear multipartite ssRNA(+)	~15.8 kb	Plants
*Polydnaviriformidae* [32]	multiple copies of segmented, circular supercoiled dsDNA	150–250 kbp	Parasitoid wasps (Ichneumonidae and Braconidae) of Lepidoptera
*Eupolintoviridae* [38]	linear dsDNA	15–40 kbp	Eukaryotic genomes
*Iridoviridae* [39]	linear dsDNA	140–303 kbp	Fish, amphibians, reptiles, insects and crustaceans
*Nudiviridae* [40]	circular dsDNA	96–232 kbp	Insects and crustaceans
*Parvoviridae* [41]	linear ssDNA	4–6 kb	Vertebrates and invertebrates

## Data Availability

Clean reads obtained from deep sequencing raw data were deposited in the National Centre for Biotechnology Information’s (NCBI’s) Sequence Read Archive (SRA) and are accessible through BioProject accession number PRJNA1008060. The virus genomic sequence obtained in the present work was deposited in the GenBank database of the National Center for Biotechnology Information (NCBI) under accession numbers (PX096626-PX096631). Further data that support the findings of this study are available from the corresponding author upon request.

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
