# Peer review of "Viral Community and Novel Viral Genomes Associated with the Sugarcane Weevil, Sphenophorus levis (Coleoptera: Curculionidae) in Brazil"

_viruses, 2025, doi:10.3390/v17101312_

Round 1

Reviewer 1 Report

Comments and Suggestions for Authors

The authors, Amanda Haisi and co-workers, presented a manuscript entitled, "Viral Community and Novel Viral Genomes Associated with the Sugarcane Weevil, Sphenophorus levis (Coleoptera: Curculionidae) in Brazil".

The authors analyzed viruses isolated from the digestive tract of Sphenophorus levis, a major pest of sugarcane in Brazil.

They chose the HTS method. They identified a number of new viruses and detected the presence of already known viruses.

Their work is important for understanding insect virus communities in general and viruses present in agricultural crop pests in particular. The authors also propose practical applications of their findings for biological control of Sphenophorus levis.

The paper is well written. The methods and procedures used are well described. The new viruses are thoroughly described, including phylogenetic analyses. The results are presented in considerable detail and compared with previous work by other authors.

Minor adjustments are needed to improve the clarity and readability of their work.

The authors refer to accompanying material, which was not available during the review.

At the end of the chapter "Results and Discussion," I recommend including a summary of which of the detected viruses have potential for biological control against Sphenophorus levis.

I also recommend specifying which specific species were detected for at least those viruses with potential practical use—the manuscript only provides identification at the family level.

Additional comments:

Line 229: Bracoviriform facetosae – do not use binomial names of viruses when the rest of the article uses classical names. Taxonomy must be used correctly and consistently.

Line 230 and other places in the manuscript: the family Adintoviridae has been recently renamed Eupolintoviridae

Line 304, caption for Figure 4: the phylogenetic tree is part marked B), not C

Author Response

Thank you very much for taking the time to review this manuscript. Please find the detailed responses below and the corresponding revisions/corrections highlighted in yellow and tracked in the re-submitted files. Point-by-point response to Comments and Suggestions for Authors:

Comments 1: The authors refer to accompanying material, which was not available during the review.

Response 1: Thank you for pointing this out. In order to facilitate the review, we inserted Table S1 (cited in lines 203, 212, 220, 225 and 460) on pages 9 and 16 of the manuscript, although it had also been mentioned in the "Supplementary Materials" section — which led to the misunderstanding. To correct this oversight, in the revised version Table S1 will be removed from the manuscript and submitted as a separate file under "Supplementary Material."

Comments 2: At the end of the chapter "Results and Discussion," I recommend including a summary of which of the detected viruses have potential for biological control against Sphenophorus levis.

Response 2: Thank you for pointing this out. This study provides the first characterization of the RNA virome associated with S. levis using a viral metagenomic approach. Although several putative viral pathogens were detected, their role and potential application as biological control agents remain to be clarified. Further research, including experimental validation and host–virus interaction studies, will be essential to assess their feasibility for integrated pest management strategies.

Comments 3: I also recommend specifying which specific species were detected for at least those viruses with potential practical use—the manuscript only provides identification at the family level.

Response 3: Thank you for pointing this out. All species specifically assigned by the Read Annotation Tool (RAT) pipeline, based on the highest similarity, are listed in Table S1 (separate file under “Supplementary Material”). We also corrected the information in Table 2 regarding the percentage of reads classified as viral. The column previously labeled “Family Level (%)” has been updated to “Family/Species Level (%)” to indicate that the reported reads include not only those classified at the family level but also unassigned reads (NA), some of which were analyzed as potential novel viral genomes in our manuscript (line 167, page 5).

Comments 4: Line 229: Bracoviriform facetosae – do not use binomial names of viruses when the rest of the article uses classical names. Taxonomy must be used correctly and consistently.

Response 4: We fully agree with the statement that taxonomy must be used correctly and consistently. In this case, however, when we wrote Bracoviriform facetosae on line 251 of page 9 and in Table S1 (separate file under "Supplementary Material.") we were reporting the result obtained using the Read Annotation Tool (RAT) pipeline, and on line 268 of the same page 9 we were citing Starchevskaya et al. (2023) exactly as it appears in that article. The species Bracoviriform facetosae is a renaming of Diolcogaster facetosa bracovirus, whose lineage has been updated. Many other insect viruses, such as those presented in the rest of the manuscript, still do not have names currently accepted by the International Committee on Taxonomy of Viruses (ICTV) (https://ictv.global/taxonomy/taxondetails?taxnode_id=202404375&taxon_name=Bracoviriform%20facetosae).

Comments 5: Line 230 and other places in the manuscript: the family Adintoviridae has been recently renamed Eupolintoviridae.

Response 5: Thank you for pointing this out. We have corrected the nomenclature of the viral family as requested (in the text: lines 252, 257, 266, 268, and 272 on page 9; in Table 3: line 192, page 8) and have added the appropriate bibliographic reference to support this modification (Reference 38).

Comments 6: Line 304, caption for Figure 4: the phylogenetic tree is part marked B), not C.

Response 6: Thank you for pointing this out, the mistake has been corrected. Now you read: "(B) The maximum likelihood phylogenetic tree of Sphenophorus levis associated virus and previously reported members of the order Picornavirales based on amino acid sequences of polyprotein" (line 327 on page 11).

Reviewer 2 Report

Comments and Suggestions for Authors

This is a meaningful study, but it still has some shortcomings.

1. After extensive descriptions of the pest’s biology and current control tactics, the authors do not explicitly state what is missing in the literature. Add one explicit sentence such as “To date, no study has characterized the RNA virome of S. levis, and thus potential viral biocontrol agents remain unidentified.” 2. Table 1 is useful, but the accompanying text is a list rather than a synthesis; it does not clarify which viral taxa are the most promising for weevil biocontrol. Conclude the paragraph with a comparative statement (e.g., “Among Curculionidae, Partitiviridae, Iflaviridae and Reoviridae are recurrent and vertically transmitted, making them prime candidates for exploitation.”) 3. The authors collected 30 adult insects and pooled them into 3 groups of 10 intestines each. However, no justification is provided for this pooling strategy or the sample size. Please clarify why 10 individuals per pool were chosen. Is this based on prior literature or pilot data? Include a power analysis or reference if available. Also, explain why whole intestines were used rather than whole bodies or specific tissues—this affects interpretation of the virome composition. 4. Suggestion: Include information on negative controls (e.g., blank extractions, mock libraries) to rule out environmental or reagent contamination. This is especially important in metagenomic studies where contamination can skew results. 5. Define the criteria for viral genome assembly, annotation, and novelty (e.g., sequence divergence thresholds, coverage requirements). Consider including genome maps, ORF predictions, and conserved domain annotations as supplementary figures. 6. Details on how insects were stored and transported from the field to the lab are missing. Include information on transport conditions (e.g., temperature, time) and how samples were preserved to ensure RNA integrity. 7. No discussion is provided on whether sequencing depth was sufficient to capture the full diversity of the virome. Include rarefaction analysis or coverage saturation plots to demonstrate that sequencing depth was adequate for viral detection.

Author Response

Thank you very much for taking the time to review this manuscript. Please find the detailed responses below and the corresponding revisions/corrections highlighted in green and tracked in the re-submitted files. Point-by-point response to Comments and Suggestions for Authors:

Comments 1: After extensive descriptions of the pest’s biology and current control tactics, the authors do not explicitly state what is missing in the literature. Add one explicit sentence such as “To date, no study has characterized the RNA virome of S. levis, and thus potential viral biocontrol agents remain unidentified.”

Response 1: We agree. Accordingly, we have revised the text to emphasize this point. The revised text now reads (lines 67–68; page 2): "In the reviewed literature, no studies were found on the RNA virome of S. levis, and thus potential viral biocontrol agents remain unidentified;"

Comments 2: Table 1 is useful, but the accompanying text is a list rather than a synthesis; it does not clarify which viral taxa are the most promising for weevil biocontrol. Conclude the paragraph with a comparative statement (e.g., “Among Curculionidae, Partitiviridae, Iflaviridae and Reoviridae are recurrent and vertically transmitted, making them prime candidates for exploitation.”)

Response 2: We agree with the reviewer that the recurrence of certain viral families in specific insect species and the possibility of vertical transmission are important indicators of potential candidates for biological control. However, given the current lack of knowledge on this subject — including basic aspects of Sphenophorus levis biology — we do not feel comfortable making such claims based on our present data. We believe this is a promising avenue that requires further investigation.

Comments 3: The authors collected 30 adult insects and pooled them into 3 groups of 10 intestines each. However, no justification is provided for this pooling strategy or the sample size. Please clarify why 10 individuals per pool were chosen. Is this based on prior literature or pilot data? Include a power analysis or reference if available. Also, explain why whole intestines were used rather than whole bodies or specific tissues—this affects interpretation of the virome composition.

Response 3: Thank you for pointing this out. The sampling in our study followed the approach of Etebari et al. (2022) with Scarabaeidae, also sugarcane pests in Australia, and according to them, the gut tissue is the most active site for viral entry and replication in other scarabs. This reference was added to the Materials and Methods section. The use of three independent pools of 10 intestines each (30 insects in total) accounts for individual variability in viral communities, while ensuring sufficient biological replication and technical feasibility. Pooling insect intestines has been widely adopted in virome studies (e.g., honey bees, termites), as it provides a representative overview of the associated virome and minimizes individual bias, while keeping sequencing costs manageable. In addition, pooling provided sufficient tissue for RNA extraction and adequate quantification for library preparation input.

The revised text now reads (lines 88- 90, page 2): "The intestines of these insects were then removed and immediately stored in RNAlater (Sigma-Aldrich), resulting in 3 pools of 10 intestines each [22].”

Comments 4: Suggestion: Include information on negative controls (e.g., blank extractions, mock libraries) to rule out environmental or reagent contamination. This is especially important in metagenomic studies where contamination can skew results.

Response 4: Thank you for the suggestion. We included a biological negative control in our library preparation and sequencing workflows by processing a pool of banana weevil (Cosmopolites sordidus) specimens in parallel. Although the data from this control were not included in this manuscript, as they are part of a separate study in preparation, the viral community detected in this control was entirely distinct from that of Sphenophorus levis. These libraries thus served to rule out environmental contamination or cross-sample carry-over.

The revised text now reads (lines 107-110, page 3): "To monitor the potential contamination during the workflow, a negative control consisting of a pool of Cosmopolites sordidus (banana weevil) specimens was processed in parallel through all steps of nucleic acid extraction, library preparation, and sequencing.

Comments 5: Define the criteria for viral genome assembly, annotation, and novelty (e.g., sequence divergence thresholds, coverage requirements). Consider including genome maps, ORF predictions, and conserved domain annotations as supplementary figures.

Response 5: We thank the reviewer for this comment and have rewritten the paragraph to incorporate the criteria for viral genome assembly, annotation, and novelty. Genome maps were generated only for the putative viral genomes, as the pipeline used for taxonomic classification performs ORF mapping against a protein database and applies a voting-based approach to classify entire contigs based on the classification of individual ORFs. To facilitate understanding, we have grouped the methodology used for viral genome analysis under section 2.4, titled Viral Genome Analysis.

The revised text now reads (lines 128-142, pages 4-5): “Based on the RAT results, contigs with the highest relative abundance—each sup-ported by more than 200,000 mapped reads and classified as “not assigned (NA)”—were selected for viral genome assembly. Contigs with minimum threshold of ≥500 nucleotides and an average coverage of ≥20x were aligned using Geneious Prime v. 2025.1.3 (Dotmatics, Boston, USA), and the longest contigs were subsequently used for molecular characterization of putative novel viral genomes. Subsequently, putative Open Reading Frames (ORFs) were identified using ORF finder (https://www.ncbi.nlm.nih.gov/orffinder/) with a length cutoff of >300 nt. The conserved domains were predicted using InterProScan (http://www.ebi.ac.uk/interpro) [28] on amino acid sequence, as well as Conserved Domain Database (CDD) v. 3.21 and Pfam v.35 on nucleotide sequences, both with a significance threshold of E-value ≤ 10⁻³. The classified contigs were subjected to BLASTx analysis against the non-redundant protein database (nr) using an e-value cutoff of <10-5 to eliminate likely false positives. Potential novel genomes were further assessed based on their genomic organization and phylogenetic topology.”

Comments 6: Details on how insects were stored and transported from the field to the lab are missing. Include information on transport conditions (e.g., temperature, time) and how samples were preserved to ensure RNA integrity.

Response 6: Thank you for pointing this out. Groups of live insects collected were placed in plastic containers (14.5 × 10.0 × 5.0 cm) with perforated lids and transported by motor vehicle under controlled conditions (25°C, complete darkness) for 3 h 30 min to the Laboratory of Entomology and Phytopathology (LEF) of Embrapa Environment (CNPMA) in Jaguariúna, SP. Upon arrival, 30 individuals were randomly selected and fasted for 72 hours in a BOD incubator maintained at 25 °C with a 24 h scotophase. To ensure RNA integrity, the intestines, immediately after removal, were preserved by immersion in RNAlater (Sigma-Aldrich) and then frozen at −20 °C (according to the manufacturer’s instructions) until processing for total RNA extraction.

The revised text now reads (lines 83-91, page 2): "Groups of live insects collected were placed in plastic containers (14.5 × 10.0 × 5.0 cm) with perforated lids and transported by motor vehicle under controlled conditions (25°C, complete darkness) for 3 h 30 min to the Laboratory of Entomology and Phytopathology (LEF) of Embrapa Environment (CNPMA) in Jaguariúna, SP. Upon arrival, 30 individuals were randomly selected and fasted for 72 hours in a BOD incubator maintained at 25 °C with a 24 h scotophase. The intestines of these insects were then removed and immedi-ately stored in RNAlater (Sigma-Aldrich), resulting in 3 pools of 10 intestines each [22]. Samples were frozen at −20 °C, following the manufacturer’s instructions, until RNA extraction."

Comments 7: No discussion is provided on whether sequencing depth was sufficient to capture the full diversity of the virome. Include rarefaction analysis or coverage saturation plots to demonstrate that sequencing depth was adequate for viral detection.

Response 7: We performed a rarefaction analysis at the viral family level based on RAT classifications, presenting the mean and standard error. This information was included in the Materials and Methods (Section 2.3), in the Results (Section 3.1), and in Supplementary Figure S1 (provided as a separate file under “Supplementary Material”).

The revised text now reads (lines 122-124, page 4): “Rarefaction curve analyses were performed using the vegan R package v.2.7.1 in R [27]. Curves were computed for each library with 100 resampling steps, and richness values (viral families) were plotted as mean ± standard error (SE).” and (lines 176-178, page 6): “As shown by the rarefaction curves of the three samples reaching the asymptote (Supplementary Figure S1), the sequencing depth was sufficient to capture the viral community at family level.”